# Evaluating the Performance of sUAS Photogrammetry with PPK Positioning for Infrastructure Mapping

Conor McMahon [1], Omar E. Mora [2],* and Michael J. Starek [3]

1    Department of Geography, University of California, Santa Barbara, CA 93106, USA;
     conor.mcmahon@geog.ucsb.edu
2    Department of Civil Engineering, California State Polytechnic University, Pomona, CA 91768, USA
3    Department of Computing Sciences, Texas A&M University-Corpus Christi, Corpus Christi, TX 78412, USA;
     michael.starek@tamucc.edu
*    Correspondence: oemora@cpp.edu

**Abstract:** Traditional acquisition methods for generating digital surface models (DSMs) of infrastructure are either low resolution and slow (total station-based methods) or expensive (LiDAR). By contrast, photogrammetric methods have recently received attention due to their ability to generate dense 3D models quickly for low cost. However, existing frameworks often utilize many manually measured control points, require a permanent RTK/PPK reference station, or yield a reconstruction accuracy too poor to be useful in many applications. In addition, the causes of inaccuracy in photogrammetric imagery are complex and sometimes not well understood. In this study, a small unmanned aerial system (sUAS) was used to rapidly image a relatively even, 1 ha ground surface. Model accuracy was investigated to determine the importance of ground control point (GCP) count and differential GNSS base station type. Results generally showed the best performance for tests using five or more GCPs or when a Continuously Operating Reference Station (CORS) was used, with vertical root mean square errors of 0.026 and 0.027 m in these cases. However, accuracy outputs generally met comparable published results in the literature, demonstrating the viability of analyses relying solely on a temporary local base with a one hour dwell time and no GCPs.

**Keywords:** sUAS; photogrammetry; mapping; DSM; PPK; GCP

## 1. Introduction

Numerous disciplines require the generation of three-dimensional models, point clouds, or maps. In civil engineering, this information is required to monitor, in real time, the dimensions of buildings [1], tunnels [2], and materials stockpiles [3] during construction, or to validate their dimensions once finished. Routine inspection of existing systems may also be required to detect possible formation of defects (deformation, spalling, or potholes) [4–7]. Mapping is also valuable for older buildings without as-built models [4,5,7], and during disaster response for damaged structures [8]. Three-dimensional models are also increasingly used in agriculture and forestry [9,10] and in the environmental sciences [9,11–21]. The required model accuracy and resolution varies substantially across applications. In forestry contexts, errors in estimated tree heights of 1 m or more are not uncommon [9]. By contrast, for some construction-monitoring applications a deviation of 0.01 m may be seen as substantial [2], while many other applications require an accuracy somewhere in between these values.

Large-scale 3D mapping projects typically utilize one of three technologies: manual total station measurements, Light Detection and Ranging (LiDAR), or photogrammetry. Total station methods are reliable and common in construction but extremely slow, requiring careful measurement of each point while moving around a bulky sensor system. By comparison, terrestrial laser scanning (TLS) systems allow the generation of millions of points

extremely quickly and with higher accuracy [22]. However, TLS instruments are expensive and often require multiple scans and subsequent labor-intensive scan-stitching routines.

Photogrammetry offers an increasingly popular alternative [15,23–33], which utilizes relatively cheap equipment—small unmanned aerial systems (sUASs) and off-the-shelf digital cameras—to produce dense point clouds much more rapidly than either of the above two methods [23]. However, these benefits historically have come at the cost of accuracy and precision in the point cloud compared to total station or LiDAR methods. The causes of inaccuracy in photogrammetric point clouds are complex [24,26–29] and scene-specific [25,28,34], and further investigation is required to elucidate the limit that these problems place on wider-scale implementation of photogrammetry in the sciences and industry.

Aerial photogrammetry using sUASs typically relies on the distribution of manually measured ground control points (GCPs) [15,23–32,34]. Recently, focus has shifted to the use of onboard high-accuracy differential GNSS to assist in the photogrammetric reconstruction. The GNSS signal from the aerial vehicle is compared to that received at a fixed, known ground base to remove a component of the location error.

A substantial amount of recent work has focused on characterizing the influence of PPK or Real-Time Kinematic (RTK) positioning and GCP use on depth model outcome. Most studies investigating GCP counts find that model results are improved by the addition of more GCPs [25] up to a certain point, beyond which the benefit diminishes [24,29,34]. The optimal number of GCPs varies, but many recent studies utilize between 12 and 20 [15,24,29,34] (although the total number will invariably depend on the size of the area to be mapped and will continue to increase for larger areas). The addition of RTK/PPK methods generally improves results over use of GCPs alone [27,29,30,32], and use of only RTK/PPK in the place of GCPs may provide results comparable to or improved over GCP use alone [26,27,29,30]. In cases where both RTK/PPK and GCPs are utilized, introduction of RTK/PPK may reduce the number of GCPs required for accuracy to reach its peak value [29]. In some cases, this removes the dependency of accuracy on GCP count [25], producing high-accuracy surveys when only a single GCP is used [31,32].

Absolute accuracy values vary substantially based on the study area and methods used. Vertical error is generally higher than horizontal error [26,29], with a few exceptions [29]. Tomaštik et al. compared model error between setups with 4 or 9 GCPs or when using PPK, and found that PPK outperformed either of their GCP approaches by large margins [26] with vertical root mean squared error (RMSE) across the checkpoints of 0.138 m. Lucieer et al. produced a digital surface model (DSM) based on 12 GCPs and simple (non-differential) GNSS with an elevation RMSE of 0.044 m [15]. In 2016, Gerke and Przybilla's lowest vertical RMSE was 0.046 m, using RTK in conjunction with four GCPs spread over a 66 ha space. In [29], twelve GCPs were required in a 2.25 ha space to achieve between 0.01 and 0.02 m ground elevation accuracy. The addition of PPK brought the number of required GCPs to yield that same result down to six. Benassi et al. were able to produce 0.02 m RMSE vertical accuracy on one flight using RTK without GCPs, but other surveys of the same site using the same pipeline had RMSEs varying up to 0.1 m [27]. In 2018, Mora utilized a combination of five GCPs and RTK to produce a vertical RMSE of 0.02 m [23]. Similarly, Bolkas was able to achieve a vertical RMSE of 0.021 m using dense GCPs and 0.055 m using PPK alone; the author also notes that inclusion of a single GCP alongside PPK helped to reduce bias in the result [29].

Several authors have investigated the interaction of other factors with GCP count and output accuracy. One recent study compared results for several different types of ground surfaces and found that accuracy was generally better for structurally simple, flat surfaces such as parking lots [28]. The authors also achieved better accuracy for a lower-flying survey at 45 m compared to a higher survey at 90 m. In another study, the authors note that in some cases model accuracy declines with distance to the nearest tiepoint [34]. Lower surveys allow denser imaging of the target surface and likely lead to denser tiepoint distributions. The authors also investigated reductions in accuracy

associated with deviations of surface angle away from nadir, areas with few overlapping images, and shadows.

Results from these prior studies show that the drivers of photogrammetric model error are complex and further study is still needed to determine best practices across use cases.

*Contributions*

As noted above, a great deal of variation exists in the literature regarding the required number of GCPs to achieve adequate depth measurement results. This study included repeat photogrammetric analyses using the same survey data but relying on different numbers of GCPs and alternately including or discluding PPK.

In addition to testing performance with and without PPK, the influence of PPK ground station was also investigated. Several Continuously Operating Reference Stations (CORSs) were used. In addition, a temporary local base station was set up on site, localized using one hour of observations, and used for PPK analysis, as well. CORS are unevenly distributed over the United States and similar networks do not exist in some countries, so the use of a local base with a short dwell time may be very attractive for rapid surveys of new locations. Notably, the dwell time required for this local base was much shorter than the time required to establish local references and measure GCPs with the total station.

Finally, the depth estimation accuracy for the best CORS base and the best GCP approaches provide a baseline level of vertical accuracy that is comparable to many of the better-performing results in the literature for similar flat surfaces [23,28,29], providing support for the idea that either GCP- or PPK-based photogrammetry surveys are a viable means of assessing infrastructure surfaces.

Thus, the primary contributions of the present study are three-fold: further investigation of the required number of GCPs for constraining sUAS photogrammetric models and evaluation of the relative merits of GCP-based and PPK-based analyses; comparison of the product quality using various fixed GNSS bases for PPK; and evaluation of the limits on possible accuracy in open infrastructure.

## 2. Materials and Methods

### 2.1. Study Area and Reference Data

The study area of approximately 1 ha is located northeast of Los Angeles, California, USA, in the city of Monrovia (Latitude: 34°08′32.34″ N, Longitude: 117°59′19.24″ W). The study area is a mostly-flat parking lot that varies in ground elevation by 8 m. It is paved, except for parked vehicles, a few parking lot islands with small trees, and light fixtures distributed throughout. Initially, two reference points were measured for two hours using a GeoMax Zenith 35 Pro GNSS antenna. The GNSS antenna was set to have a position dilution of precision (PDOP) mask of 4.0, an elevation mask of 15° and a logging interval of 1 s. Subsequently, the two reference points were processed using the National Oceanic and Atmospheric Administration (NOAA) Online Positioning User Service (OPUS). The accuracy level of the two reference points was 0.02 m for both the horizontal and vertical components. The horizontal coordinate system was the North American Datum of 1983 (NAD 83), State Plane Coordinate System, California Zone V. The vertical coordinate system was the North American Vertical Datum of 1988 (NAVD 88). Once the two reference points coordinates were computed, a robotic total station survey was performed to establish the coordinates of the GCPs using a GeoMax Zoom90 total station, GeoMax 360 degree robotic prism, and a GeoMax PS336 data collector. The total station had an angular accuracy of 1 s and a distance accuracy of 1 mm ± 1.5 ppm. Field measurements using the Robotic Total Station were made to establish the 3D coordinates of the GCPs—a total of 29 GCPs were established. One of the two reference points was used to measure all the ground points, and the other reference was backsighted. The 29 GCPs provide complete coverage of the study area and a sufficient sample size for the evaluation. All GCPs measured were corners on the parking lot striping. Of the ground points measured, nine were selected for used as GCPs for further analysis, with the other 20 functioning as checkpoints (see Table 1). The

number and identity of the checkpoints was held constant through all tests. Figure 1 shows the study area, including the distribution of the GCPs tested and the two reference points.

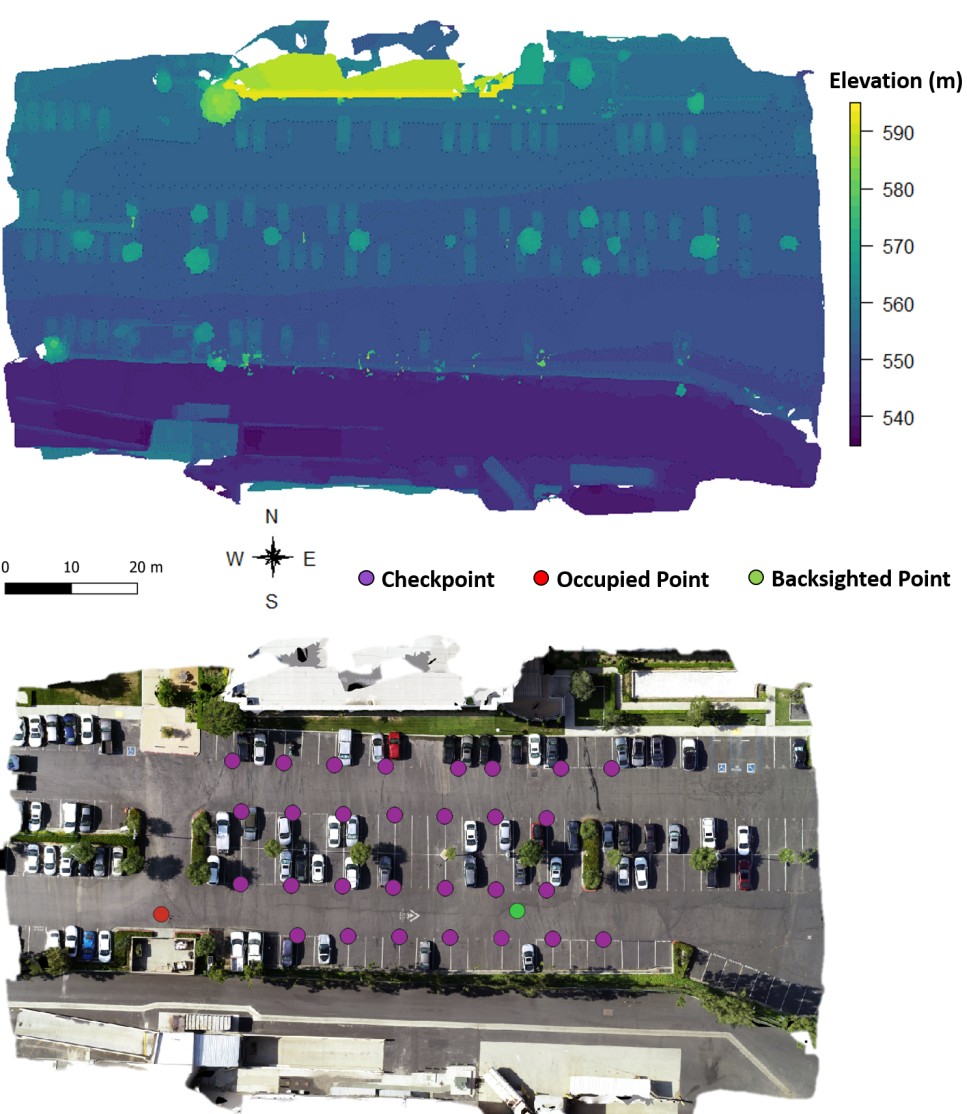

**Figure 1.** Digital surface model (**above**) and orthophoto (**below**) of the study site. Reference point locations (GCPs and checkpoints) are shown in purple on the orthophoto image.

**Table 1.** Input constraints for photogrammetric pipeline. Thirteen iterations were performed with different GCP and PPK fixed base options. Seven of these only utilized GCPs without access to PPK information, while four utilized only PPK without reliance on GCPs. Two cases used a small number of GCPs along with PPK. GCPs were always drawn from the same pool of 9 candidate points, with the other 20 points acting as checkpoints for all studies. Even in cases where fewer than 9 GCPs were used, the remaining points were not added to the checkpoint count to ensure that the checkpoints used were constant across analyses.

| Treatment | GCP Count | Checkpoint Count | PPK Base |
|---|---|---|---|
| 1 GCP + PPK | 1 | 20 | Local |
| 2 GCPs + PPK | 2 | 20 | Local |
| 3 GCPs | 3 | 20 | none |
| 4 GCPs | 4 | 20 | none |
| 5 GCPs | 5 | 20 | none |
| 6 GCPs | 6 | 20 | none |

**Table 1.** *Cont.*

| Treatment | GCP Count | Checkpoint Count | PPK Base |
|---|---|---|---|
| 7 GCPs | 7 | 20 | none |
| 8 GCPs | 8 | 20 | none |
| 9 GCPs | 9 | 20 | none |
| JPLM | 0 | 20 | JPLM |
| AZU1 | 0 | 20 | AZU1 |
| VDCY | 0 | 20 | VDCY |
| Local | 0 | 20 | Local |

### 2.2. sUAS Image Acquisition

The aerial survey was performed on 9 August 2019, using a DJI Inspire 2 sUAS with a Zenmuse X4S 20 megapixel camera with GNSS. Prior to PPK correction, the DJI Inspire 2 GNSS specifications list an expected accuracy at ±0.33 ft vertically and ±0.98 ft horizontally. The sUAS flew at about 30 m above the ground collecting images at nadir with a forward overlap of 80% and a sidelap of 80% and an average ground sample distance (GSD) of 8.5 mm. The flight speed was 2.2 m/s, the temperature was 24° Celsius, and the imagery was acquired around 10:30 a.m. local time. The drone flew for 5 min and 30 s and captured 80 images. The DroneDeploy application [35] was used during image collection, which constructed an autopilot flight path based on input parameters, including flying height, forward lap, and sidelap. The base station used for the GNSS-PPK survey was placed on an aforementioned reference point. The base station collected observations for 1 h and was placed roughly 20 m away from the test site, where there was an unobstructed line of sight between the sUAS and base station. The base station data collection followed the same framework as the reference point data with the same equipment.

### 2.3. Postprocess Kinematic (PPK) Direct Geopositioning

The sUAS was equipped with a third generation Loki system from GeoCue. The system includes a GNSS and Postprocess Kinematic (PPK) direct geopositioning system for sUAS. Direct Geopositioning is a method in which a high-accuracy positioning system is flown on a sUAS to determine camera position estimates. The advantage of using a direct geopositioning approach is that it improves the horizontal and vertical accuracy of sUAS mapping, while reducing the number of GCPs required. The Loki system estimates a priori X, Y, and Z positions for each image after postprocessing the data in ASPSuite. Upon processing the imagery in ASPSuite, a 0.02 m accuracy was achieved for all images in both the horizontal and vertical components. ASPSuite processes all imagery in NAD 83 in the National Spatial Reference System (NSRS) of 2011 for the horizontal and NAVD 88 for the vertical coordinate systems.

### 2.4. Continuously Operating Reference Stations (CORSs)

The National Geodetic Survey (NGS) manages a network of CORSs that provide GNSS data in support of three-dimensional positioning. Coordinates enhanced by postprocessing with CORS approach a few centimeters relative to the NSRS, both horizontally and vertically. CORS data are downloaded from a nearby station to supply image corrections. These corrections are used during postprocessing in ASPSuite to accurately and precisely geotag images. The CORS used in our evaluation were three nearest to our study area: Jet Propulsion Lab Mesa (JPLM) operated by the NASA Jet Propulsion Laboratory and VDCY and AZU1 operated by UNAVCO-PBO. Detailed information on the three stations is provided in Table 2.

**Table 2.** Specifications for the three CORS used for PPK in this study.

| Station | Sample Rate | GNSS | Distance (km) | Location |
|---|---|---|---|---|
| JPLM | 1 s | GPS + GLO | 18.36 | 34°12′17.34″ N, 118°10′23.57″ W |
| AZU1 | 15 s | GPS | 8.69 | 34°07′33.66″ N, 117°53′47.31″ W |
| VDCY | 15 s | GPS | 21.65 | 34°10′42.82″ N, 118°13′11.95″ W |

### 2.5. sUAS Image Processing

The sUAS image dataset was processed using the software ContextCapture [36]. The general framework of ContextCapture includes automatic aerial triangulation (AAT), bundle block adjustment (BBA), point cloud, DSM, and orthophoto creation. The sUAS manufacturer delivered approximations for the interior orientations parameters that were adjusted during the data processing using a self-calibration. All photogrammetric processing was performed using an Intel® Xeon® CPU ES-2620 v4 @ 2.10 GHz (2 processors) with 128 GB RAM, 2 NVIDIA Quadro P2000 graphics card, and 36 physical cores.

### 2.6. Surface Model and Orthophoto Generation

The stitched orthophoto from the routine using 8 GCPs is presented in Figure 1, with 8.54 mm ground resolution. The total ground area imaged is 8089 m², with about 111 million pixels recovered. Figure 1 also provides the analogous height surface model, at the same spatial resolution and extent. The surface heights in this region varied from 533 to 593 m, with most of the scene dominated by a parking lot, associated cars, and vegetation. The ground height of the parking lot surface varied from 548 to 556 m, with all of the checkpoint heights falling within this band.

### 2.7. sUAS and Robotic Total Station Comparison

The 3D position of the robotic total station-derived coordinates were examined against the sUAS image derived positions by manually identifying the checkpoints in the sUAS images following the same process used for associating GCPs in ContextCapture.

### 2.8. Tests of GCP and PPK Influence on Accuracy

In order to investigate the influence of GCP count and PPK usage on output model accuracy, the ContextCapture pipeline was run thirteen times with different input information. For seven iterations, no PPK information was utilized and the camera calibration and depth model generation was run using only the photography and manually labelled GCP correspondences. These seven approaches each used different numbers of GCPs, ranging from 3 to 9, with 20 of the other manually measured points acting as checkpoints. The individual checkpoints used as GCPs are displayed in Figure 2. In four other approaches, no GCPs were used and the 20 non-GCP points were used as checkpoints. In the next four cases, the model was built using PPK and the fixed base used to localize the moving sUAS was varied. Three of these cases used CORSs—JPLM, AZU1, and VDCY. See Table 2 for more information on these CORSs. The fourth PPK approach used a temporary local base set up on a known reference point at the beginning of the survey. The local base was situated less than 30 m from the study site and was recorded concurrently with the sUAS survey for one hour. In two final routines, 1 and 2 GCPs were used in combination with PPK using the local base. This information is also presented in Table 1.

After running the ContextCapture pipeline with each of the above settings, the results were analyzed to determine the influence of GCP count and PPK usage on survey and processing time, tiepoint production, reprojection error, and checkpoint location error (Section 3). Error at each individual checkpoint across all processing methods was also plotted to investigate the possibility that some checkpoints were more difficult to localize than others. These and other results are detailed throughout Section 3 below.

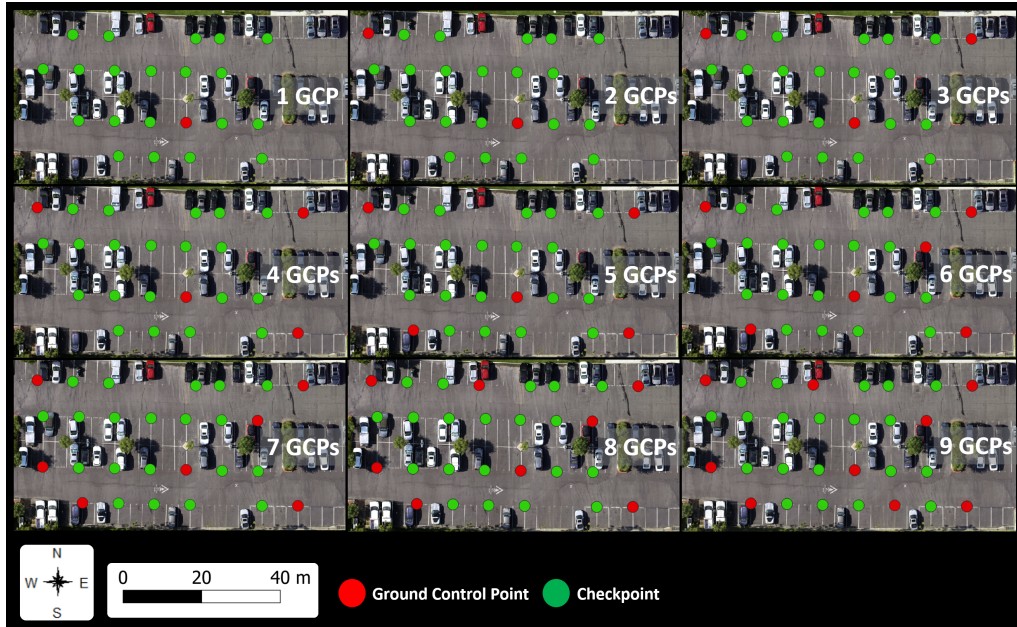

**Figure 2.** Map showing the location of selected GCPs for each test as red circles. All test configurations are shown, with GCP counts ranging from 1 to 9. GCPs were selected from the 29 checkpoints to minimize the distance between any point in the surveyed area and its nearest GCP. For each routine the points used as GCPs are drawn from a single pool of 9 ground points, while the remaining 20 points were used as checkpoints.

## 3. Results

### 3.1. Camera Calibration

The automatic camera calibration results from each method are provided in Table 3. The methods using PPK all produced very similar estimates for the six camera parameters, while the methods utilizing only GCPs yielded a different set of mutually similar values. The focal length predictions from GCP methods were much closer to the default manufacturer specification of 8.8 mm when compared to the PPK results, which were generally about 13% higher. Additionally, the three $k_i$ parameter terms which allow for correction of radial distortion were more negative for the PPK approaches when compared to the GCP approaches. By comparison, there was a less clear difference in the tangential distortion correction terms ($p_1$, $p_2$) between approaches; these correction terms were uniformly very small, which may imply that most of the variation there was noise.

**Table 3.** Automatic camera calibration results from ContextCapture photogrammetry routine. $K_1$, $K_2$, and $K_3$ are the radial correction terms while $P_1$ and $P_2$ are tangential correction terms. The average ground pixel size and offset in principal point position in X and Y are also provided.

| Treatment (Units) | Focal Length (mm) | $K_1$ - | $K_2$ - | $K_3$ - | $P_1$ - | $P_2$ - | Pixel Size (mm) | $PP_x$ | $PP_y$ |
|---|---|---|---|---|---|---|---|---|---|
| Local | 10.02 | 0.0109 | −0.0324 | 0.0341 | 0.0012 | 0.00020 | 8.54 | 2449 | 1845 |
| JPLM | 10.02 | 0.0107 | −0.0323 | 0.0339 | 0.0012 | 0.00020 | 8.54 | 2449 | 1845 |
| AZUI | 9.98 | 0.0109 | −0.0324 | 0.0335 | 0.0012 | 0.00020 | 8.54 | 2449 | 1845 |
| VDCY | 10.00 | 0.0111 | −0.0344 | 0.0358 | 0.0012 | 0.00020 | 8.55 | 2449 | 1845 |
| 1 GCP + PPK | 10.01 | 0.0105 | −0.0306 | 0.0319 | 0.0012 | 0.00024 | 8.52 | 2449 | 1845 |
| 2 GCPs + PPK | 10.00 | 0.0106 | −0.0314 | 0.0328 | 0.0011 | 0.00022 | 8.53 | 2449 | 1845 |
| 3 GCPs | 8.80 | 0.0061 | −0.0170 | 0.0134 | 0.0010 | 0.00005 | 8.54 | 2435 | 1832 |
| 4 GCPs | 8.80 | 0.0066 | −0.0160 | 0.0126 | 0.0010 | 0.00001 | 8.54 | 2435 | 1831 |
| 5 GCPs | 8.80 | 0.0068 | −0.0169 | 0.0132 | 0.0010 | −0.00006 | 8.54 | 2434 | 1830 |
| 6 GCPs | 8.80 | 0.0071 | −0.0175 | 0.0137 | 0.0010 | −0.00009 | 8.54 | 2434 | 1830 |
| 7 GCPs | 8.80 | 0.0070 | −0.0175 | 0.0137 | 0.0009 | −0.00013 | 8.54 | 2434 | 1829 |
| 8 GCPs | 8.80 | 0.0090 | −0.0217 | 0.0169 | 0.0010 | −0.00012 | 8.54 | 2434 | 1833 |
| 9 GCPs | 8.80 | 0.0089 | −0.0216 | 0.0169 | 0.0009 | −0.00014 | 8.54 | 2434 | 1833 |

*3.2. Survey and Processing Time Costs*

Combined survey and postprocessing time costs varied across the survey methods tested; these are presented in Table 4. GCP-based methods with more GCPs took much longer total times due to the need to manually survey more points. Additionally, most PPK methods required more photogrammetry processing time in ContextCapture than most GCP-based methods, although there was considerable variation in PPK time cost and this was a small component of overall survey time regardless. The PPK methods also required time for the postflight correction of image locations using PPK, which required about 5.45 times as much overall time as the flight itself, and about 15.3 times as much time as the average ContextCapture routine.

ContextCapture processing time for all methods was small compared to the manual checkpoint location survey time of two hours. As the PPK-based methods were not dependent on GCP field measurements, the total time cost in the field for the methods not using GCPs was only 30.7% of that for the fastest method based exclusively on GCPs (3 GCPs), or 12.9% of the field time cost for the slowest GCP-based method (9 GCPs). This decrease in field campaign time requirements is balanced against an increase in automated postprocessing time in an office environment for the PPK itself. Combining PPK with a small number of GCPs may provide an attractive balance between these constraints. For more details on the hardware utilized, see Section 2.5.

**Table 4.** Survey and processing time costs for photogrammetry routines in seconds. Survey time is given as the total two-hour time block required to survey all 29 ground points using the total station, reduced by a factor equal to the proportion of all 29 GCPs required for each algorithm. The PPK approaches did not utilize GCPs and so this total station survey time is not included for PPK algorithms. Only one flight was conducted and used in all postprocessing approaches, so flight time was the same for all cases. Total time is the sum of flight, survey, and postprocessing time.

| Treatment | Total Time | Field Labor | Flight | Survey | Photogrammetry | PPK |
|---|---|---|---|---|---|---|
| Local | 2230 | 330 | 330 | 0 | 100 | 1800 |
| JPLM | 2222 | 330 | 330 | 0 | 92 | 1800 |
| AZU1 | 2242 | 330 | 330 | 0 | 112 | 1800 |
| VDCY | 2272 | 330 | 330 | 0 | 142 | 1800 |
| 1 GCP + PPK | 2511 | 578 | 330 | 248 | 133 | 1800 |
| 2 GCPs + PPK | 2739 | 827 | 330 | 497 | 112 | 1800 |
| 3 GCPs | 1190 | 1075 | 330 | 745 | 115 | 0 |
| 4 GCPs | 1441 | 1323 | 330 | 993 | 118 | 0 |
| 5 GCPs | 1691 | 1571 | 330 | 1241 | 120 | 0 |
| 6 GCPs | 1940 | 1820 | 330 | 1490 | 120 | 0 |
| 7 GCPs | 2189 | 2068 | 330 | 1738 | 121 | 0 |
| 8 GCPs | 2438 | 2316 | 330 | 1986 | 122 | 0 |
| 9 GCPs | 2685 | 2564 | 330 | 2234 | 121 | 0 |

*3.3. Tiepoint Distribution*

All approaches generated the same number of keypoints within the images: 18,991. From these keypoints, tiepoints were generated as matches between images. In ContextCapture, the entire aerotriangulation (including constraints from GCPs, photo geodata, and tiepoint generation) was performed all in one step and therefore each test can generate a different number and distribution of tiepoints. Here, the median number of tiepoints recovered per image varied from 1294 to 1514; exact quantities are presented in Table 5. Tiepoints were unevenly distributed across the scene, with dense cover in areas of high visual texture (e.g., cars, painted lines on edges of parking spaces) and fewer points in regions with less texture (e.g., bare flat pavement). The spatial distribution of tiepoints is illustrated in Figure 3.

**Table 5.** Tiepoint production for each of the different methods. The last column shows median tiepoint production across all photos.

| Treatment | Keypoints | Tiepoints | Tiepoints per Photo |
|---|---|---|---|
| Local | 18,991 | 20,385 | 1348 |
| JPLM | 18,991 | 20,356 | 1352 |
| AZU1 | 18,991 | 19,582 | 1294 |
| VDCY | 18,991 | 19,661 | 1318 |
| 1 GCP + PPK | 18,991 | 20,621 | 1427 |
| 2 GCPs + PPK | 18,991 | 19,223 | 1344 |
| 3 GCPs | 18,991 | 21,598 | 1500 |
| 4 GCPs | 18,991 | 21,629 | 1499 |
| 5 GCPs | 18,991 | 21,587 | 1495 |
| 6 GCPs | 18,991 | 21,582 | 1496 |
| 7 GCPs | 18,991 | 21,552 | 1487 |
| 8 GCPs | 18,991 | 21,694 | 1510 |
| 9 GCPs | 18,991 | 21,674 | 1514 |

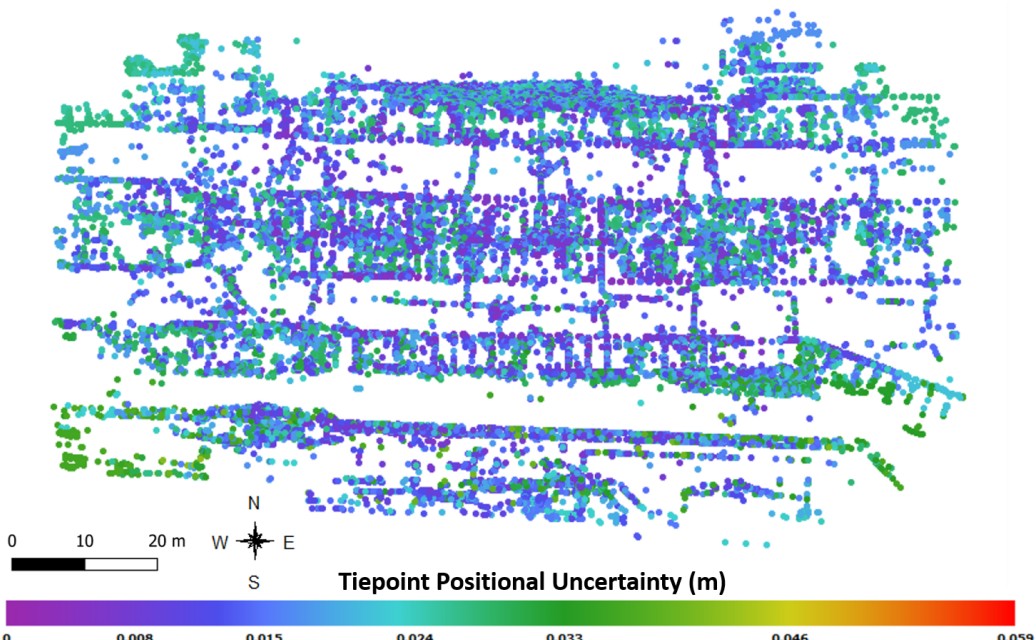

**Figure 3.** Distribution of tiepoints over the surveyed area. Tiepoint color indicates positional uncertainty in meters. Tiepoint spatial distributions were visually similar for all methods tested, although exact tiepoint counts and locations varied.

This work did not attempt to quantify errors in dense point cloud products and focused only on checkpoint error. Other works have shown that point cloud error may be exacerbated within the point cloud further from tiepoints or GCPs [34]. The uneven distribution of tiepoints through the scene (Figure 3) would likely result in increased error at some locations lacking in tiepoints. More tiepoints were generated along the edges of islands, trees, cars, and paint marks on the lot surface. It is likely that local error in the dense point cloud would be higher in flat, black areas of parking lots with low visual texture.

### 3.4. Reprojection Error

The reprojection error on manually delineated checkpoints across the images was evaluated for each treatment group—the resulting error distributions are presented in Figure 4. The GCP-based methods achieved lower reprojection error values than the four PPK methods, and use of the VDCY and local PPK bases resulted in higher error than the JPLM or AZU1 bases. Addition of one or two GCPs to the local base solution improved reprojection error but did not achieve errors as low as for the many-GCP solutions.

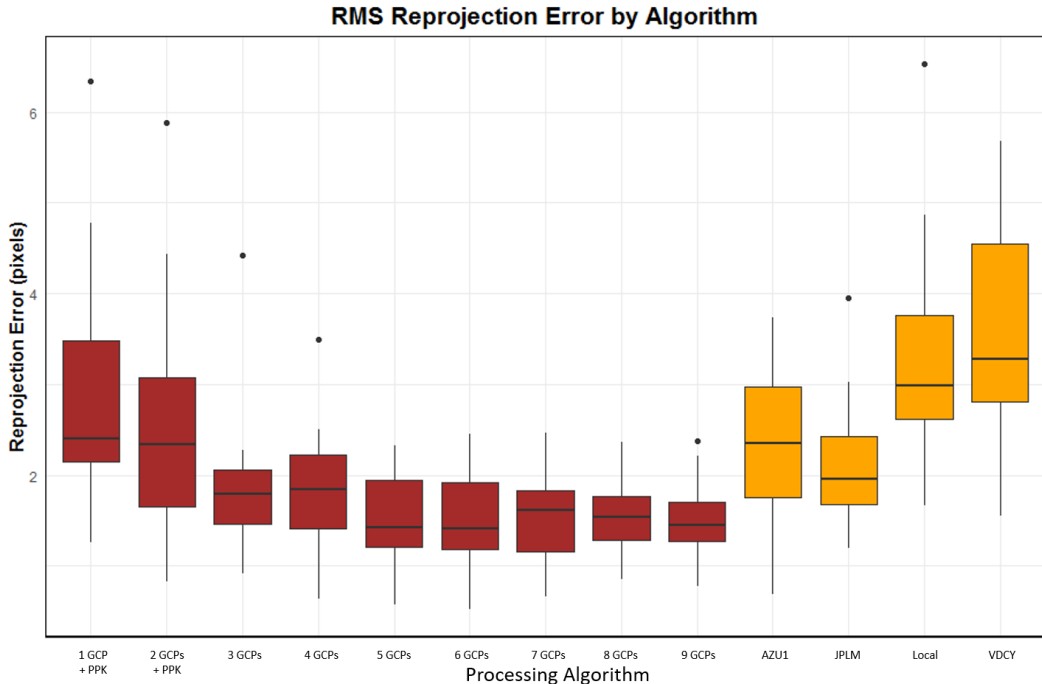

**Figure 4.** RMS reprojection error distributions across all checkpoints for each processing method. Boxplot notches are based on a confidence interval with width $\frac{1.58 IQR}{\sqrt{n}}$ around the median, where IQR is the interquartile range and n is the number of checkpoints used in that treatment.

### 3.5. Checkpoint Location Error

Checkpoint location error distributions in the horizontal and vertical directions were compared across the various processing treatments (see Figure 5). The error at each checkpoint within each test is also visualized spatially within Figure 6. The root mean square error values in both directions were also evaluated, and are presented in Table 6. The four PPK methods yielded generally higher error values than the seven GCP-based methods. JPLM, the one PPK approach using a CORS with a 1 s sampling frequency, achieved lower error values than the other two CORS methods, which relied on survey stations with 15 s sampling frequencies (see Table 2). This base also uses both GPS and GLONASS satellites, meaning it has access to a total constellation of 55 satellites. The other bases only utilized GPS, which has 31 satellites. For JPLM, the average vertical accuracy was approximately on par with the vertical accuracy values for the seven GCP tests, although the horizontal accuracy was degraded. The solutions using both local base PPK and 1 or 2 GCPs improved error compared to when only the local base was used, especially in the vertical dimension, which is consistent with other results in the literature [30,31]. The reduction in horizontal error was less. Consequently, these combined approaches yielded much better vertical accuracy than any of the zero-GCP approaches other than JPLM, with similar horizontal error to those approaches.

### 3.6. Error Differences Across Checkpoints

Error distributions were also compiled for each checkpoint across all methods, to investigate the possibility that some points were intrinsically more difficult to localize than others. Figure 7 provides graphical error distributions for each checkpoint in the vertical and horizontal directions. A few points appear especially challenging to localize compared to the rest (e.g., point 1028), but there does not appear to be any spatial explanation for this effect. Additionally, there is no clear study-wide correlation between vertical and horizontal error at each given checkpoint. Figure 8 depicts the error at each checkpoint based on the distance to the nearest GCP. In this study, GCP distance was not found to strongly constrain checkpoint location error.

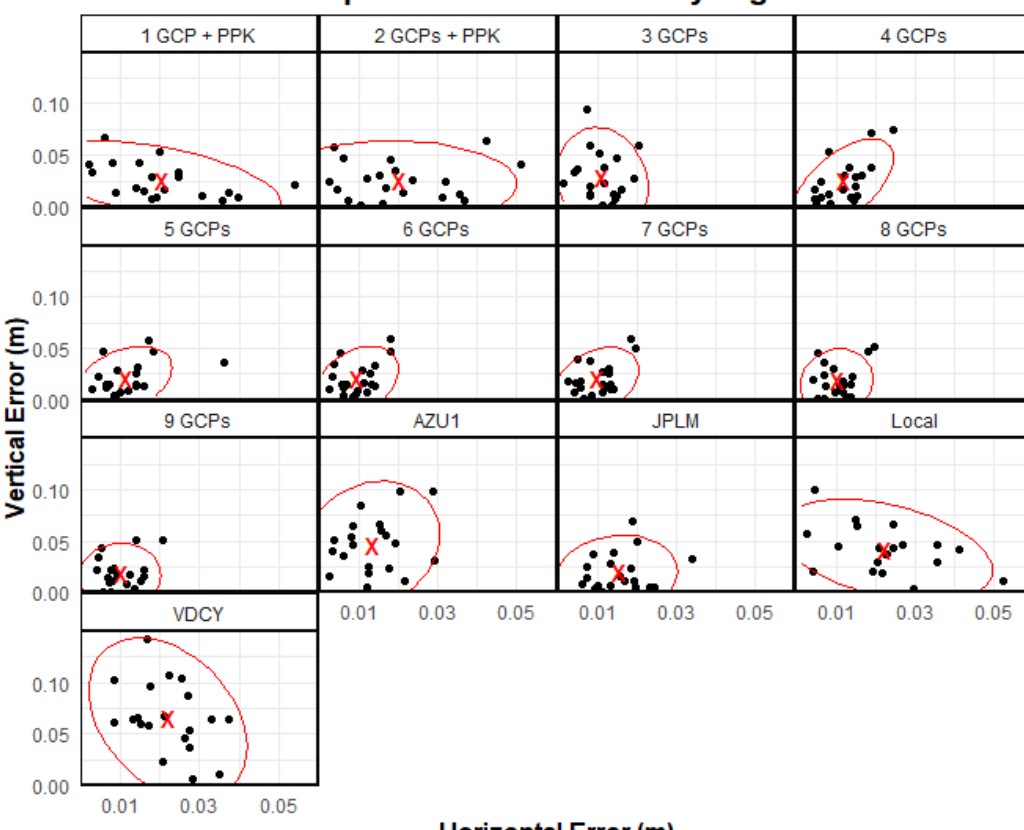

**Figure 5.** Horizontal and vertical error magnitude for each checkpoint, split by algorithm type. The mean errors across all checkpoints are presented as red X marks. Red ellipses illustrate the bounds of 95% of the variation around the mean, assuming a bivariate normal distribution with major axes aligned to the highest principal component of measurement error.

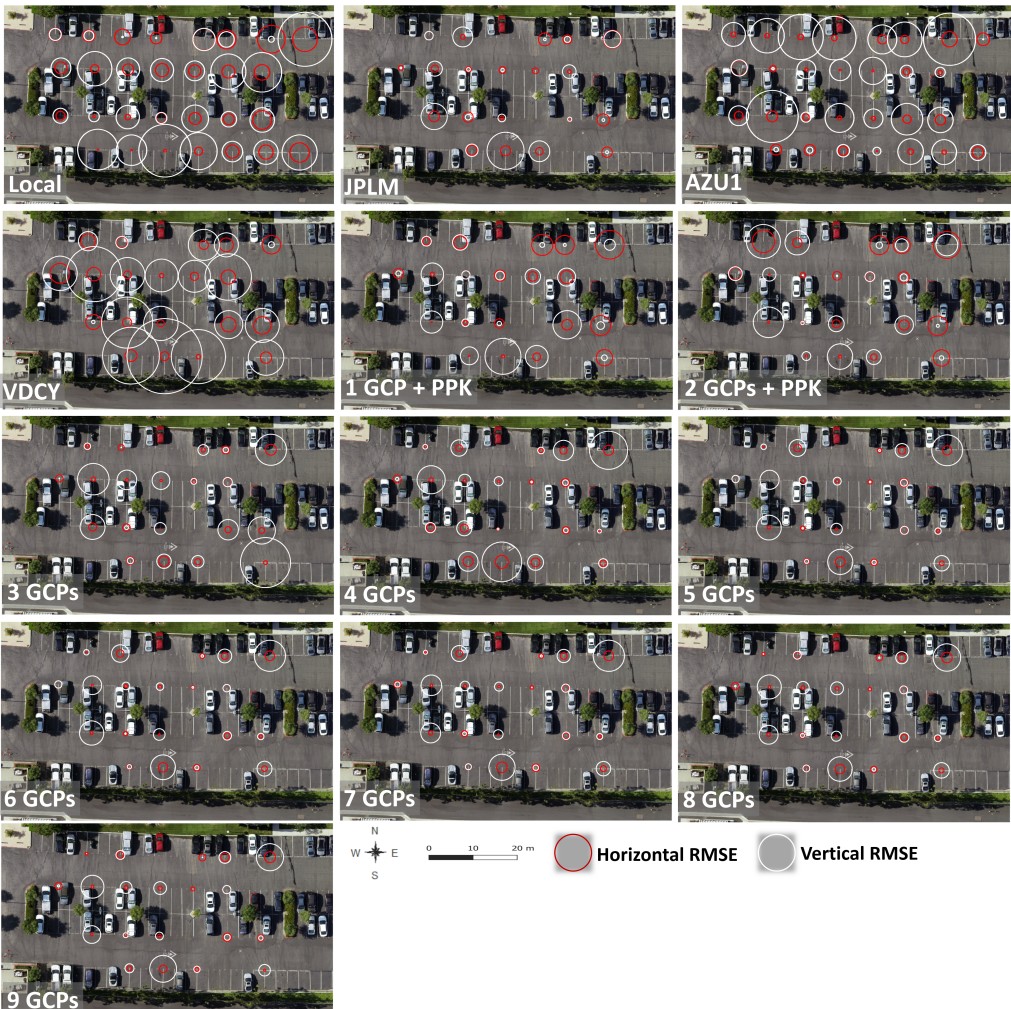

**Figure 6.** Absolute value error in checkpoint location for each method. White circles indicate vertical error for each checkpoint, while red circles give the horizontal error. Circle radius is equal to the error value, at 200 times scale for visibility (1 mm error = 2 m radius).

**Table 6.** Root mean square error and mean error values, or bias, in meters across all checkpoints for each processing algorithm.

| Treatment | RMSE (m) | | | Bias (m) | | |
|---|---|---|---|---|---|---|
| | Horizontal | Vertical | 3D | Horizontal | Vertical | 3D |
| Local | 0.026 | 0.047 | 0.054 | 0.023 | −0.039 | 0.045 |
| JPLM | 0.017 | 0.027 | 0.032 | 0.016 | −0.003 | 0.016 |
| AZU1 | 0.016 | 0.054 | 0.056 | 0.014 | 0.044 | 0.046 |
| VDCY | 0.024 | 0.074 | 0.077 | 0.022 | −0.066 | 0.070 |
| 1 GCP + PPK | 0.025 | 0.031 | 0.039 | 0.021 | −0.011 | 0.023 |
| 2 GCPs + PPK | 0.024 | 0.031 | 0.040 | 0.020 | 0.005 | 0.021 |
| 3 GCPs | 0.012 | 0.037 | 0.039 | 0.011 | 0.006 | 0.013 |
| 4 GCPs | 0.013 | 0.033 | 0.035 | 0.012 | −0.003 | 0.012 |
| 5 GCPs | 0.014 | 0.026 | 0.028 | 0.012 | 0.006 | 0.013 |
| 6 GCPs | 0.011 | 0.026 | 0.028 | 0.010 | 0.007 | 0.012 |
| 7 GCPs | 0.011 | 0.026 | 0.028 | 0.010 | 0.004 | 0.011 |
| 8 GCPs | 0.011 | 0.025 | 0.027 | 0.010 | 0.001 | 0.010 |
| 9 GCPs | 0.011 | 0.024 | 0.027 | 0.010 | −0.001 | 0.010 |

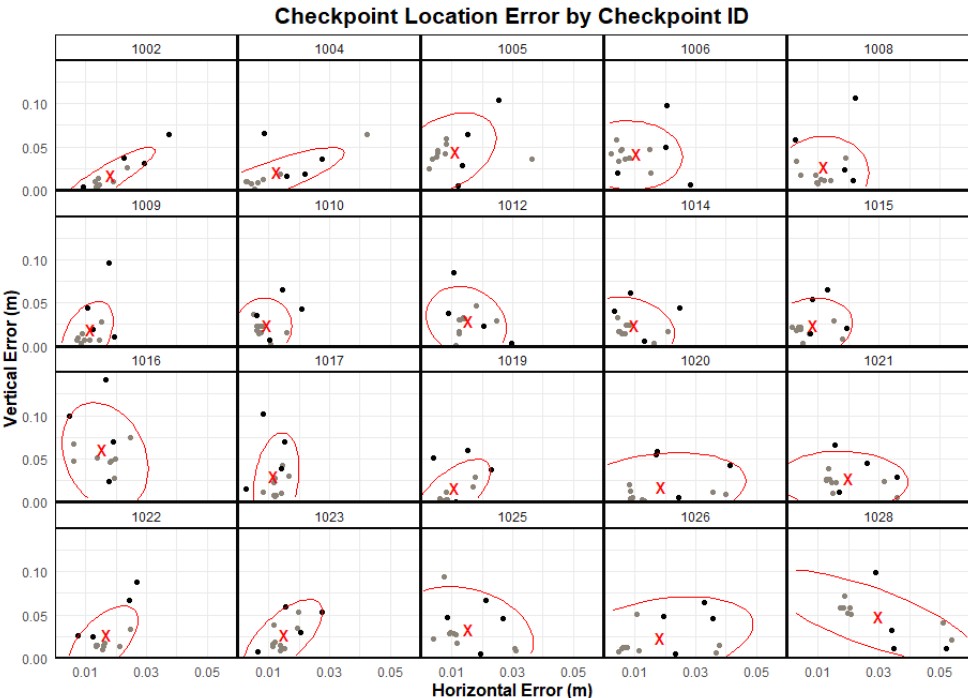

**Figure 7.** Horizontal and vertical error magnitudes for each algorithm, split by checkpoint ID. PPK results are given in black, while GCP results are in gray. The mean errors across all treatments for a given algorithm are presented as red X marks. Red ellipses illustrate the bounds of 95% of the variation around the mean, assuming a bivariate normal distribution with major axes aligned to the highest principal component of measurement error.

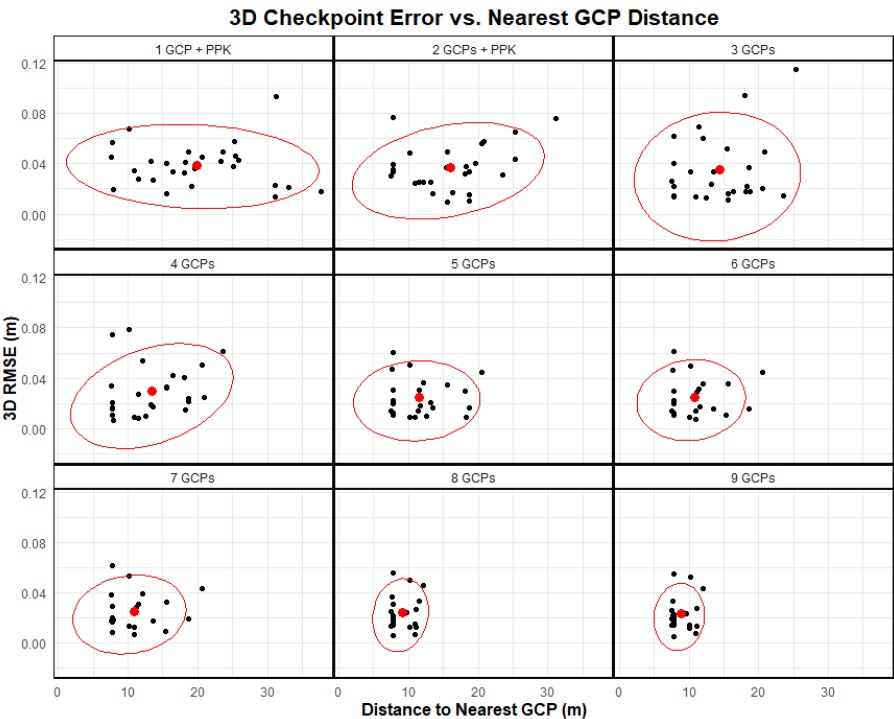

**Figure 8.** 3D RMSE in checkpoint location estimate plotted against the distance from that checkpoint to the nearest GCP. Data are split by GCP number used in the various treatments. Red ellipses illustrate the bounds of 95% of the variation around the mean, assuming a bivariate normal distribution with major axes aligned to the highest principal component of measurement error.

## 4. Discussion

Much recent work on photogrammetry has focused on evaluating the accuracy of 3D outputs at checkpoints. RMSE values are frequently assessed across all ground-measured checkpoints and used as a benchmark for overall map accuracy. The RMSE values achieved here for the nine GCP solution provide similar or better results compared to most studies in the literature on similar ground surfaces, with horizontal and vertical values of 0.011 and 0.024 m, respectively. These RMSE values are approximately 1.3 and 2.8 times the ground sample distance.

By comparison, the PPK methods produced less accurate outputs in terms of both checkpoint RMSE and reprojection error. However, these approaches do not depend on costly field measurements of GCPs using a total station, which makes the survey much faster to perform (see Table 4). The JPLM PPK error results are comparable to many other GCP-based results in the literature, while requiring only 12.9% as much total field survey time as the nine GCP method. PPK methods offer substantial potential to improve the speed at which photogrammetry systems can be deployed when used as a replacement for GCPs. This expands the feasibility of photogrammetry to a greater range of tasks in industry and science. The difference may also be magnified for larger scenes, which require more GCPs to survey. Additionally, PPK allows surveys to retain accuracy when extended outside the bounds of the control network established by a GCP survey.

Among the four PPK bases tested, the JPLM method performed the best by most metrics, including checkpoint location RMSE and reprojection error. This base has a much faster sampling frequency than the other two CORSs, at 1 s instead of 15 s. Between the two CORS with 15 s sampling frequencies, the VDCY base was about 2.5 times as far away as the AZU1 base (21.65 and 8.69 km, respectively). This difference in distance to the test site may have been responsible for the difference in performance between the two methods, with the closer base providing higher accuracy in every metric than the further site. Most of these average or RMSE values were degraded by approximately 50% for the VDCY base when compared to the AZU1 base (e.g., 0.074 and 0.054 m vertical RMSE, respectively).

The local base performed similarly to the middle-performing CORS (AZU1), which is also extremely encouraging for cases where no nearby CORS system is available. Although the GCP and CORS PPK methods generally yielded lower error values, the 0.05 m vertical RMSE value achieved using the local base may be acceptable for a wide variety of coarse mapping tasks where extremely high-accuracy height models are not necessary. Additionally, inclusion of either one or two GCPs along with the local base PPK substantially improved vertical RMSE and bias. Combining a small number of GCPs with local base PPK may be an attractive approach to provide quick and low-cost surveys with fairly high accuracy. Additionally, the local base used here had a relatively short dwell time by industry standards. This is attractive for rapid surveys in new areas, but it is possible that with a longer dwell time accuracy could have been even higher, and more similar to the JPLM results.

Checkpoint RMSE performance among methods using only GCPs was the worst for the setup with only three GCPs, followed by four GCPs. However, beyond this point the RMSE in checkpoint position appeared to stabilize, with no obvious reduction in the RMSE for higher GCP counts. This general trend corresponds to other results in the literature, wherein increases in GCP count sometimes improve performance only asymptotically up to a saturation point. In many other studies, the number of GCPs used to reach this point is higher—from 8 to 20 [24,29]. A result indicating a decrease in the number of required GCPs is encouraging, as GCP mensuration in the field is the most time-inefficient and costly part of the photogrammetry pipeline. However, it is difficult to make direct cross-study comparisons of this sort as the requirements for GCP count may change depending on the surface type measured [29] or the total survey area, with larger and more complex areas generally requiring more GCPs. This survey was conducted on a relatively flat and small asphalt parking lot, and this may have contributed to the low number of GCPs required to achieve a high-accuracy model.

The tests with PPK estimated different camera calibration parameters than the tests with only GCPs did, but as neither approach yielded consistently better checkpoint RMSEs for all cases it is not clear which set of calibration parameters is closer to the true values. In particular, the two surveys with both PPK and one or two GCPs had dissimilar camera calibration parameters to the three GCP survey, but similar 3D error. However, the focal length values produced by the GCP solutions were closer to the manufacturer specifications, and the distortion parameter lower. It is possible that differences in camera calibration may have greater effects for more complicated scenes where the checkpoints are not all situated on a roughly planar surface.

For all methods tested, the vertical error values are consistently higher than the horizontal error values (see Figures 5 and 6 and Table 6). This is also consistent with other published works [23,25,26,29], with only a few exceptions where vertical and horizontal accuracy were similar [15]. On each given survey, some checkpoints exhibited much higher errors than did others (Figure 6), but these differences were not consistent across survey methodologies (Figure 7). Additionally, there was no obvious spatial relationship explaining checkpoint error, with high-error checkpoints distributed seemingly at random throughout the scene, and there was similarly no obvious relationship between checkpoint accuracy and distance to the nearest GCP (Figure 8). There also does not seem to be an association between horizontal and vertical error values achieved at a given checkpoint within a given treatment (i.e., vertical and horizontal error values are not linearly correlated in Figure 5), implying that for a given point, errors in the horizontal and vertical directions are independent. This lack of a relationship may not hold for other settings with meaningfully sloped surfaces, where horizontal error can lead to vertical error as well.

The actual error values achieved with most GCP-based methods were approximately 0.011 and 0.025 m RMSE in the horizontal and vertical directions, respectively. These results are similar to or better than most contemporary reports in the literature [15,24–26], in particular in terms of the vertical RMSE value. For contemporary cases with similar or lower error values reported, generally either more GCPs were required [27,29] or GCPs were used in combination with PPK [23,27,29].

Vertical error values achieved using only PPK with the high sampling-frequency and GPS + GLONASS CORS were similar to those from the GCP surveys, with error values in the horizontal plane being elevated. The error resulting from use of a local base was meaningfully higher than that when using GCPs or the JPLM base, at RMSE values of 0.026 and 0.051 m in the horizontal and vertical directions. However, this level of error is still similar to contemporary works on similar surfaces, and would be acceptable for many survey tasks not dependent on extremely high-precision results. The error was also substantially lessened for cases using only 1 or 2 GCPs in combination with a local PPK base.

However, direct comparisons between studies are challenging due to differences in site size, surface complexity, and flight height. In general, lower flights and simpler surfaces result in better accuracy values [28], and this survey was of a flat surface from a relatively low altitude compared to most other published works (30 m). Consequently, comparisons to other studies should be made only in this context.

## 5. Conclusions

The error metrics achieved here using either four GCPs or the JPLM base perform at least as well as other contemporary works, with RMSE values suitable for a wide range of monitoring purposes, including disaster response, structural damage assessment, and forestry and agricultural surveys. These results show that high-accuracy surveys are possible using either fewer GCP measurements (especially for small survey areas) or none at all (when PPK is used); this is especially significant as GCP localization represents the bulk of photogrammetric operational cost. As well, demonstration of the viability of PPK using a temporary local base instead of a commercial CORS is extremely attractive for any

work occurring at remote sites without CORS availability, or where CORS usage bears commercial fees.

Other key findings of this work include an apparent decrease in output model error when using PPK with CORS when the base either is closer or has a higher sample rate—with a larger improvement here associated with the improved sample rate. Further work should be targeted at investigating these details so that best practices can be created for CORS selection in cases where multiple bases are available nearby a study site. Additionally, it would be useful to investigate the impact of parameters related to the local base—such as dwell time and distance from the study site—on model accuracy outcomes.

Future work could focus more on how the geometric distribution of GCPs influences model accuracy, and how study area size constrains the number of GCPs required for high-accuracy results. Additional sources of error worth investigating include flight height [29], surface type to be measured [29], and required output image pixel density. Finally, while some authors have carried out work to assess the causes and severity of inaccuracies in dense point clouds away from GCPs [34], much work remains to be conducted in that area. Additionally, only one actual survey flight was performed. Performing more repeated surveys of the same or different target surfaces to ensure that random differences between individual surveys are well resolved would provide us with useful information.

As high-quality camera, GNSS, and drone equipment become increasingly robust and low-cost, it is likely that error values will continue to fall even farther. Further study of the methodological influences on error in sUAS photogrammetry will continue to be required through this period and into the future.

**Author Contributions:** Conceptualization, C.M. and O.E.M.; methodology, O.E.M.; software, O.E.M. and C.M.; validation, O.E.M.; formal analysis, C.M.; investigation, C.M., O.E.M. and M.J.S.; resources, O.E.M. and M.J.S.; data curation, C.M. and O.E.M.; writing—original draft preparation, C.M. and O.E.M.; writing—review and editing, C.M., O.E.M. and M.J.S.; visualization, C.M.; supervision, O.E.M. and M.J.S.; project administration, O.E.M. All authors have read and agreed to the published version of the manuscript.

**Funding:** This research received no external funding.

**Institutional Review Board Statement:** Not applicable.

**Informed Consent Statement:** Not applicable.

**Data Availability Statement:** Data and scripts pertaining to this work are available at https://github.com/conormcmahon/photogrammetry_testing_cc, accessed on 23 May 2021.

**Acknowledgments:** The authors thank Land Design Consultants, Inc. for helping with the data collection for this study.

**Conflicts of Interest:** The authors declare no conflict of interest.

## Abbreviations

The following abbreviations are used in this manuscript:

| PPK | Postprocessing Kinematic |
| GCP | Ground Control Point |
| CORS | Continuously Operating Reference Station |

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
