# Peer review of "Evaluating the Performance of sUAS Photogrammetry with PPK Positioning for Infrastructure Mapping"

_drones, doi:10.3390/drones5020050_

Round 1
Reviewer 1 Report
The manuscript presents Evaluating the Performance of sUAS Photogrammetry With PPK Positioning For Infrastructure Mapping
In general, I appreciate the work that has been done and presented here. The topic is of interest to improve UAS studies, but in my view, the study must be improved for reasons below.
1- The introductory background is too lengthy that the focus and contributions of this study is not clearly articulated.
2- In the introduction, all text except for lines 115-117 refers to generic background information about UAS, photogrammetry, georefering, etc.
3- Many general descriptions on the metodology in section 2 also feel distracting.
4- Tables and figures must be improved.
In my review, I have included numerous suggestions. The authors need to improve their work, mention background information only when they are necessary or at least relevant for this study, and also rewrite the introductory part of the paper to clearly state the motivation and contribution of this study. I think the paper can be publishable with these significant changes.

Author Response
The reviewer's style and grammatical comments on the manuscript have all been implemented (see attached .pdf with point-by-point responses).
Additional responses to the reviewer's other comments are given here:
1- The introductory background is too lengthy that the focus and contributions of this study is not clearly articulated.
- Some extraneous text has been removed from the introduction, especially comparative discussion of survey methodologies not investigated here (e.g. TLS). The first paragraph, which includes information on motivating applications for photogrammetry and other 3D modelling techniques, has also been substantially shortened.
- Several other paragraphs were also shortened slightly to increase clarity. Overall, the introduction is now about 22 lines shorter.
- The final subsection of the introduction includes an explicit statement of this study's contributions. That subsection has been restructured somewhat for clarity. A sentence summarizing the contributions has been moved to the end of this section as well.
2- In the introduction, all text except for lines 115-117 refers to generic background information about UAS, photogrammetry, georefering, etc.
- Lines 115-117 are a brief summary of the previous paragraphs (lines 61-114 in the original manuscript) which together show that error in photogrammetric models is complex and continued research is required to understand it. Most of this information comes from previous studies looking at the influence GCP quantity and RTK/PPK parameters on output accuracy. Those influences are the focus of this study. We feel that the information summarized here is vital to understanding the current work and its context in the literature. However, some text concerning general function of UAS and PPK has been removed.
3- Many general descriptions on the metodology in section 2 also feel distracting.
- We agree that the methods section is dry, but feel that this information is critical to understand the exact methods used. Small changes in the way reference points are determined or UAS are operated can have meaningful impacts on model accuracy. We felt it was important to provide all relevant information to ensure that the results shown here can be taken in appropriate context.
- For example, flight height is important to interpret absolute RMSE values because flights from lower heights tend to have lower RMSE (e.g. Bolkas 2019).
4- Tables and figures must be improved.
- There were errors in the LaTeX formatting of the original manuscript which caused several figures and tables to be improperly placed and improperly numbered. These have been corrected.
We thank the reviewer for their insight and thoughtful commentary!
Reviewer 2 Report
The article deals with 3 themes: investigation
of the required number of GCPs for constraining sUAS photogrammetric models and evaluation of the relative merits of GCP-based and PPK-based analysis; comparison of the product quality using different fixed GNSS bases for PPK; and evaluation of the limits on possible accuracy in open infrastructure.
I agree that PPK (post-processing kinematic) solutions offer the substantial potential to improve the speed of photogrammetry systems. The article gives an interesting look at this issue.
Image acquisition and processing, surface model, and orthophoto generation are correct. In the article, it would be necessary to consider not only the use of naturally signaled points but especially the use of code targets with automatic detection of the center of the mark. But I respect that the authors have decided not to use this opportunity.
Comments:
Rows 480 - 498 must be deleted
Author Response
The reviewer notes that: "In the article, it would be necessary to consider not only the use of naturally signaled points but especially the use of code targets with automatic detection of the center of the mark. But I respect that the authors have decided not to use this opportunity."
- We believe this point is addressed at lines 301-308, which state that this study primarily focused on error at checkpoints (target locations on the ground like those the reviewer suggests) and did not attempt to quantify error at tiepoints or throughout the scene away from checkpoints. The latter is an interesting question and an important area of ongoing work, but was not a focus of this study.
The reviewer also commented that lines 480-492 should be deleted.
- These lines include the contributions statement required by MDPI. However, there were some oversights with contributions that were not properly filled out with author initials. Those corrections have been made here to remove extraneous contribution information which was not relevant for this study.
We thank the reviewer for their insight and thoughtful commentary.